# Contribution of Non-Coding RNAs to Anticancer Effects of Dietary Polyphenols: Chlorogenic Acid, Curcumin, Epigallocatechin-3-Gallate, Genistein, Quercetin and Resveratrol

**DOI:** 10.3390/antiox11122352

**Published:** 2022-11-28

**Authors:** Sumio Hayakawa, Tomokazu Ohishi, Yumiko Oishi, Mamoru Isemura, Noriyuki Miyoshi

**Affiliations:** 1Department of Biochemistry and Molecular Biology, Graduate School of Medicine, Nippon Medical School, Bunkyo-ku, Tokyo 113-8602, Japan; 2Institute of Microbial Chemistry (BIKAKEN), Numazu, Microbial Chemistry Research Foundation, Shizuoka 410-0301, Japan; 3Institute of Microbial Chemistry (BIKAKEN), Laboratory of Oncology, Microbial Chemistry Research Foundation, Shinagawa-ku, Tokyo 141-0021, Japan; 4Tea Science Center, University of Shizuoka, Shizuoka 422-8526, Japan; 5Graduate School of Integrated Pharmaceutical and Nutritional Sciences, University of Shizuoka, Shizuoka 422-8526, Japan

**Keywords:** dietary polyphenols, anticancer, ROS, noncoding RNAs, microRNA, long noncoding RNA, circular RNA

## Abstract

Growing evidence has been accumulated to show the anticancer effects of daily consumption of polyphenols. These dietary polyphenols include chlorogenic acid, curcumin, epigallocatechin-3-*O*-gallate, genistein, quercetin, and resveratrol. These polyphenols have similar chemical and biological properties in that they can act as antioxidants and exert the anticancer effects via cell signaling pathways involving their reactive oxygen species (ROS)-scavenging activity. These polyphenols may also act as pro-oxidants under certain conditions, especially at high concentrations. Epigenetic modifications, including dysregulation of noncoding RNAs (ncRNAs) such as microRNAs, long noncoding RNAs, and circular RNAs are now known to be involved in the anticancer effects of polyphenols. These polyphenols can modulate the expression/activity of the component molecules in ROS-scavenger-triggered anticancer pathways (RSTAPs) by increasing the expression of tumor-suppressive ncRNAs and decreasing the expression of oncogenic ncRNAs in general. Multiple ncRNAs are similarly modulated by multiple polyphenols. Many of the targets of ncRNAs affected by these polyphenols are components of RSTAPs. Therefore, ncRNA modulation may enhance the anticancer effects of polyphenols via RSTAPs in an additive or synergistic manner, although other mechanisms may be operating as well.

## 1. Introduction

A number of epidemiological studies have provided evidence for the anticancer effects of daily polyphenol intake [1,2,3,4,5,6]. These dietary polyphenols include chlorogenic acid (CGA), curcumin (CUR), epigallocatechin-3-*O*-gallate (EGCG), genistein (GEN), quercetin (QUE) and resveratrol (RES), which are six major polyphenols in our dietary life and are found in vegetables, fruits, and beverages. Preclinical and cell-based studies have supported their anticancer effects and provided a mechanism of action for these polyphenols [1,2,3,5]. Recent studies have shown involvement of epigenetic modifications including dysregulation of noncoding RNAs (ncRNAs) such as micro RNAs (miRs), long noncoding RNAs (lncRs), and circular RNAs (circRs).

We have provided updated information from human studies supporting the anticancer effects of consumption of green tea, coffee, red wine, soybeans, and curry and discussed the involvement of miRs in polyphenol action mechanisms (Table 1 and Table 2). Previous data have shown that the six major dietary polyphenols, CGA, CUR, EGCG, GEN, QUE, and RES, have similar properties since they can act as antioxidants and exert the anticancer effects via reactive oxygen species (ROS)-scavenger-triggered anticancer pathways (RSTAPs) (Figure 1) [7,8]. These dietary polyphenols are also known to act as pro-oxidants and ROS generated can activate AMP-activated protein kinase (AMPK) which will result in polyphenols’ anticancer effects (Figure 1). Moreover, data indicated that at least three of the six polyphenols can commonly modulate several miRs associated with RSTAPs [7,8].

In this review, we further discuss the miR-modulating effects of polyphenols, which have been reported in studies using one or two of the six dietary polyphenols (Table 3 and Table 4). Furthermore, based on recent evidence on involvement of lncRs and circRs in anticancer mechanisms of these polyphenols, we summarize the modulatory effects of six dietary polyphenols on lncRs and circRs in relation to their anticancer effects. 

## 2. Involvements of miRs in Polyphenol-Mediated Anticancer Mechanisms

miRs are defined as small single-stranded molecules (approximately 20 to 25 nucleotides) and can regulate gene expression at the transcriptional and post-transcriptional levels, leading to modulation of beneficial health effects exerted by these polyphenols in diseases including cancer [7,8].

Table 1 and Table 2 summarize miRs modulated by at least three of six dietary polyphenols. Four of six dietary polyphenols upregulate miR-16, 34a and 141, and downregulate miR-20a and 221; five of six dietary polyphenols upregulate miR-145 and downregulate miR-21 and 155. Table 1 and Table 2 also list the molecular targets of miRs that are modulated by these polyphenols; targets associated with RSTAPs (Figure 1) are also shown in these tables. Thus, it appears that six dietary polyphenols can exert their anticancer effects not only by directly involving RSTAPs, but also by miR-mediated regulation of the molecular targets associated with RSTAPs.

One or two of the miRs up- and down-regulated by six polyphenols for which studies have been reported are listed in Table 3 and Table 4, respectively, together with determined or proposed targets of these miRs. Many miRs can target components of RSTAPs, but some contribute to other mechanisms that are not depicted in these pathways (Figure 1). Based on previous findings on positive crosstalk between NF-κB and Wnt/β-catenin signaling [144,145], the Wnt/β-catenin signaling is connected in Figure 1. Furthermore, previous findings are incorporated to show that TNF-α activates Wnt/β-catenin pathway, leading to increases in cancer stemness and epithelial-to-mesenchymal transition (EMT) which are involved in cancer cell renewals and tumorigenesis [146,147,148].

## 3. Involvements of lncRs in RSTAPs

LncRs are more than 200 bases long and are species/tissue specific. LncRs can interact with DNA, RNA, and proteins to regulate a wide range of biological processes [149,150]. Recent microarray analyses and next-generation sequencing assays have found that lncRs can modulate gene expression at multiple levels, including regulation at the epigenetic, transcriptional, and post-transcriptional levels. In addition, lncRs can act as sponges or molecular decoys for miRs, and reduce a population of valid specific miR, and thus, influence miR activity.

There are two major categories of lncRs, which are defined as oncogenic lncRs such as MALAT1, HOTAIR, SOX2-OT and H19, and tumor suppressing lncRs such as MEG3, PANDAR, GAS5, and TUG1 according to their pathological features [151]. Evidence is accumulating that dietary polyphenols including six polyphenols can modulate lncRs. Table 4 shows the modulations of lncRs by six polyphenols. The possible contribution of these lncRs to RSTAPs is shown in Figure 1.

### 3.1. lncR Modulations by CGA


**
*GAS5*
**


For reasons unknown, information on CGA’s actions on lncRs is quite limited. In a study on CGA’s preventive effects on *Salmonella typhimurium*-induced intestinal diseases, Tan et al. [152] found that CGA prevented pathological damages and upregulated intestinal lncR GAS5. Cellular experiments showed that GAS5 competitively bound oncogenic miR-23a to reduce the population of miR-23a as a molecular sponge, resulting in upregulation of phosphatase and PTEN. Since the increased expression of PTEN is implicated in RSTAPs (Figure 1), this finding could be applied to explain the anticancer activity of CGA, although this needs to be validated using cancer cells.


**
*TUG1*
**


In a study on a role of CGA in protection against oxidative stress injury in glaucoma, Gong et al. [153] found that CGA promoted Nrf2 expression via upregulation of lncR TUG1 in retinal ganglion RGC-5 cells and decreased cell apoptosis. Based on this finding one may expect that CGA exerts anticancer effects via its upregulation of TUG1, since antioxidative natural phytochemicals have been reported to prevent carcinogenesis by upregulation of Nrf2 [154,155].

### 3.2. lncR Modulations by CUR


**
*AK294004*
**


In nasopharyngeal carcinoma (NPC) cell lines (CNE-2), a microarray analysis showed that expression of a number of lncRs was changed by X-ray irradiation and that the expression of 116 ncRNAs was restored by CUR [156]. The results of qRT-PCR confirmed these changes in six lncRs (AF086415, AK095147, RP1-179N16.3, MUDENG, AK056098 and AK294004). Functional studies suggested that cyclin D1 is a direct target of AK294004. Radiotherapy is one of the most effective treatment modalities for NPC patients, and radioresistance is the main risk factor contributing to poor prognosis. This resistance occurs with the first X-ray treatment and the survived cells become more resistant to the second X-ray treatment, leading to invalidation of the treatment. Thus, CUR can be expected to improve radiosensitivity by altering expression of these lncRs.


**
*GAS5*
**


Dendrosomal CUR treatment of MCF7, MDA-MB231 and SKBR3 cells increased the expression of Tusc7 and GAS5 [157]. GAS5 downregulation suppressed many anticancer effects of dendrosomal CUR in breast cancer (BCa) cells. Because an amplified level of GAG5 has been reported to reduce chemotherapy resistance [158], Co-treatment of dendrosomal CUR with GAS5 overexpression could be a clinically useful tool against drug-resistant BCa cells.


**
*H19*
**


Kujundzić et al. [159] found that CUR inhibited cell proliferation and suppressed expression of lncR H19 in several human cancer cell lines such as HCT 116, SW 620, and HeLa. CUR’s downregulation of H19 expression was not found in primary normal thyroid cells [159]. Similarly, Liu et al. [160] found that CUR inhibited the proliferation of gastric cancer (GCa) SGC-7901 cells, suppressed H19 expression, and increased p53 expression. Ectopic expression of H19 attenuated CUR-induced apoptosis and downregulated p53 expression. CUR downregulated the expression of c-Myc oncogene and addition of c-Myc protein in the cell culture medium attenuated the CUR-induced downregulation of H19 expression, which may explain part of the anticancer mechanism of CUR.

In tamoxifen-resistant MCF-7 BCa cells, CUR decreased the expression levels of the epithelial marker E-cadherin, increased the expression levels of mesenchymal marker N-cadherin, and decreased H19 expression [161]. Overexpression of H19 induced EMT, invasion and migration by upregulating Snail, a key regulator of the EMT process. CUR attenuated H19-induced alterations in N-cadherin and E-cadherin expression levels and inhibited H19-induced invasion and migration, indicating that CUR may prevent H19-associated cancer cell metastasis.


**
*KCNQ1OT1*
**


The cisplatin-resistant colorectal cancer (CRC) HCT8/DDP cells exhibited higher expression levels of oncogenic KCNQ1OT1 compared to non-resistant cells. Zheng et al. [162] found that CUR promoted apoptosis in HCT8/DDP cells and silencing of KCNQ1OT1 enhanced apoptosis in the cisplatin-resistant cells. KCNQ1OT1 was found to eliminate the suppressive effect of miR-497 on expression of anti-apoptotic Bcl-2. KCNQ1OT1 overexpression counteracted the effect of CUR on these cells via miR-497/Bcl-2 axis. CUR downregulated KCNQ1OT1 expression, leading to suppression of cisplatin resistance. This may explain CUR’s reducing effect on cisplatin resistance.


**
*LINC00691*
**


In papillary thyroid cancer B-CPAP cells, CUR decreased cell proliferation, promoted apoptosis, and inhibited LINC00691 expression [163]. CUR administration or transfection of si-LINC00691 caused downregulation of AKT leading to apoptosis in these cells, suggesting that inhibition of LINC00691 is involved in the anticancer effect of CUR.


**
*linc-PINT*
**


Microarray experiments with acute lymphoblastic leukemia cells from the patients showed that 43 lncRs were aberrantly expressed as compared to healthy donor blood cells [164]. qRT-PCR found that 15 out of the 16 tested lncRs examined had the same expression pattern in the expression array including downregulation of linc-PINT. Overexpression of this lncR in Molt-4 cells induced the transcription of HMOX1, which reduced cell viability. CUR was found to upregulate the expression of linc-PINT and HMOX1 in Molt-4 cells, suggesting that upregulation of linc-PINT may be one of the CUR’s anticancer mechanisms.


**
*MEG3*
**


Alghanimi and Ghasemian [165] showed that dendrosomal CUR promoted cell death in BCa MCF-7 cells, increased gene expression of lncR MEG3, and decreased expression of FOXCUT gene. Since previous studies have shown that overexpression of MEG3 is associated with inhibition of cancer cell growth [166], CUR’s upregulation of this lncR may contribute to its anticancer effects. CUR induced apoptosis of gemcitabine-resistant non-small cell lung cancer (NSCLC) cell lines A549/GR and H520/GR and upregulated the expression of MEG3 and PTEN [167]. MEG3 overexpression increased PTEN expression and knockdown of MEG3 decreased its expression. MEG3 knockdown or PTEN knockdown mitigated CUR’s effects on these cells.


**
*NRB2*
**


Yu et al. [168] found that CUR upregulated lncR NBR2 in CRC cell lines including HCT116, HCT8, SW620, and SW480, and inhibited CRC cell proliferation by activating the AMPK pathway and inactivating mTOR. These effects of CUR were cancelled by knockdown of NBR2, indicating that modulation of the NBR2/AMPK/mTOR pathway may be involved in the anticancer effects of CUR.


**
*PANDAR*
**


CUR increased PANDAR expression in CRC DLD-1 cells and silencing of PANDAR increased apoptosis and attenuated cell senescence by stimulating expression of PUMA [169]. Knockdown of PANDAR switched CUR-induced senescence to apoptosis, suggesting the usefulness of CUR in CRC therapy. Since PUMA has been reported to initiate apoptosis by dissociating Bax and Bcl-X(L), leading to activation of proapoptotic function of Bax [170], it may be speculated that CUR-mediated upregulation of PANDAR may involve in the anticancer effect of CUR via Bax activation.


**
*PVT1*
**


The EZH2, a subunit of polycomb repressive complex 2 (PRC2), is known to have an important role in drug resistance. EZH2 interacts with several lncRs including PVT1 to modulate EMT and cancer stemness related to drug resistance. Yoshida et al. [171] found that CUR sensitized chemoresistant cancer cells by inhibiting the expression of EZH2 and PVT1 using gemcitabine-resistant pancreatic ductal adenocarcinoma cells, suggesting that CUR can overcome chemoresistance in pancreatic ductal adenocarcinoma patients via inhibition of the PRC2-PVT1-c-Myc axis.


**
*ROR*
**


Shao et al. [172] demonstrated that CUR inhibited the cell growth of human hepatoma SMMC7721 and Huh-7 cells through inducing cell cycle arrest and apoptosis and downregulated ROR expression. Overexpression of ROR restored CUR-induced growth inhibition and inactivated Wnt/β-catenin signaling, suggesting that downregulation of ROR is involved in the anticancer effects of CUR.

In CD44^+^/CD133^+^ human prostate cancer (PCa) stem cells derived from the PCa cell lines Du145 and 22RV1, CUR treatment resulted in the inhibition of cell growth and invasion, and cell cycle arrest along with decreased expression of stem cell marker proteins such as Oct4 [37]. In addition, high miR-145 expression and suppression of ROR expression were observed in the CUR-treated cells. Bioinformatic analysis and luciferase activity assays showed that Oct4 and ROR directly compete for miR-145 binding. Thus, CUR’s anticancer activity involves its downregulation of ROR which functions as miR sponge to competitively bind tumor suppressing miR-145, contributing to an increase in the population of miR-145, resulting in downregulation of Oct4, which plays a critical role in cancer development and progression [8,173].


**
*UCA1*
**


In A549 cells, CUR inhibited cell proliferation and cyclin D1 expression, and enhanced cell apoptosis [174]. CUR inhibited UCA1 expression, leading to downregulation of Wnt/mTOR pathway. Overexpression of UCA1 attenuated the effect of CUR on apoptosis. Based on previous findings that knockdown of UCA1 reduces c-Myc expression [175], CUR’s anticancer effect may be associated with downregulation of c-Myc via downregulation of UCA1.


**
*XIST*
**


Sun et al. [176] found that XIST was downregulated in renal cell carcinoma (RCC) tissues and cells such as ACHN, Caki-1, Caki-2, and 786-O. Overexpression of XIST suppressed cell proliferation, induced cell cycle arrest at G0/G1 in cultured cells, and inhibited tumor growth in a xenograft model. XIST directly interacted with miR-106b-5p and increased p21 expression. CUR regulated XIST/miR-106b-5p/p21 axis in RCC cells, indicating a role of XIST in RCC.

### 3.3. lncR Modulations by EGCG


**
*AF085935*
**


Sabry et al. [177] showed that the combination of EGCG and metformin was highly effective against the proliferation of hepatoma HepG2 cells. This combination downregulated AF085935 and glypican-3 and promoted apoptosis via upregulation of caspase 3 and downregulation of survivin. However, the direct target of AF085935 has not yet been identified.


**
*LINC00511*
**


Zhao et al. [111] found that EGCG modulated the expression of a number of lncRs in GCa AGS and SGC7901 cells. EGCG suppressed oncogenic LINC00511 and knockdown of LINC00511 inhibited cell proliferation and promoted cell death. LINC00511 could decrease the expression of miR-29b, followed by inducing GCa development. Knockdown of miR-29b rescued the effects of LINC00511 silencing. Overexpression of KDM2A, a target of miR-29b, restored the level of LINC00511.


**
*NEAT1*
**


CTR1 is known to promote cisplatin internalization in tumor cells. Jiang et al. [135] found that EGCG induced CTR1 and enhanced cisplatin sensitivity in NSCLC cells. miR-98-5p suppressed CTR1 gene expression, while NEAT1 enhanced it. Bioinformatics analysis showed that miR-98-5p is a target of CTR1. NEAT1 can be a competing endogenous lncR that upregulates EGCG-induced CTR1 by sponging miR-98-5p in these cells, suggesting that EGCG is an effective chemotherapeutic agent in the lung cancer treatment. Similarly, Chen et al. [178] found that EGCG increased ROS levels, expression of CTR1 and NEAT1 in tumor tissue, and suppressed ERK1/2 and p-ERK1/2 in a nude mouse xenografts model of lung cancer.

Cancer stem cells have been implicated as a major player in tumor metastasis, tumor recurrence, and chemotherapy resistance. CTR1 is associated with cisplatin resistance. Jiang et al. [179] found that in cancer stem cell-rich cells derived from parent lung cancer NSCLC cells, NEAT 1 was upregulated and CTR1 was downregulated. EGCG downregulated NEAT1 and suppressed the stemness triggered by overexpressing NEAT1 via inducing CTR1 expression. Wnt signaling pathway and EMT process were shown to be involved in NEAT1-induced cancer cell stemness in NSCLC.


**
*SOX2OT variant 7*
**


Wang et al. [180] found the synergistic effect of EGCG with an antitumor drug doxorubicin on osteosarcoma cells. EGCG targeted SOX2OT variant 7 via Notch3 signaling pathway and decreased stemness including drug resistance, tumorigenic ability, and self-renewal ability of these cells.


**
*Other studies*
**


The results of lncR microarray analysis revealed that EGCG treatment of lung cancer cells caused significant alterations in a total of 960 lncRs and 1434 mRNAs [181]. Among them, upregulation of five lncRs (ENSG00000272796.1, ENSG00000254054.2, ENSG00000260630.2; SNAI3-AS1, ENSG00000235142.2; LINC0532 and ENSG00000224063.1; CALCRL-AS1) and downregulation of five lncRs (ENSG00000251018.2, ENSG00000226403.1, PSMC3IP, ENSG00000230109.1 and SG00000130600.10) were confirmed by qRT-PCR. Bioinformatic analysis suggests that potential anticancer mechanisms by which EGCG regulates lncRs are associated with RSTAP members such as AKT1, caspase 3, and p53 and others, but the individual targets of these lncRs remain to be determined.

### 3.4. lncR Modulations by GEN


**
*HOTAIR*
**


In PCa cells, PC3 and DU145, GEN downregulated oncogenic HOTAIR and upregulated miR-34a [29]. Luciferase reporter assays showed that miR-34a bound to HOTAIR, leading to downregulation of HOTAIR. Knockdown of HOTAIR by siRNA caused inhibition of PCa cell growth, migration and invasion and induced apoptosis, indicating that GEN exerts its anticancer effects via downregulation of HOTAIR, which is also targeted by miR-34a. Similarly, Chiyomaru et al. [35] reported that GEN downregulated HOTAIR and upregulated miR-141 which bound to HOTAIR, leading to suppression of HOTAIR in RCC 786-O and ACHN cells. GEN inhibited proliferation and induced apoptosis in BCa MCF-7 cells. GEN decreased AKT phosphorylation and the expression of its downstream target, HOTAIR [182].

Imai-Sumida et al. [183] showed that GEN inhibited the interaction between HOTAIR and PRC2 in RCC 786-O and ACHN cells, leading to tumor suppression. GEN upregulated the tight junction protein ZO-1 by reducing the recruitment of PRC2 to the ZO-1 promoter. GEN also inhibited interaction between HOTAIR and SMARCB1, one of subunits of the human chromatin remodeling complex. These findings suggest that suppression of HOTAIR/chromatin remodeling pathways is involved in GEN’s anticancer effects against these cancer cells. Other studies have demonstrated that HOTAIR promotes cancer cell proliferation by activating the PI3K/AKT/mTOR signaling pathway [184,185], suggesting that GEN exerts its anticancer effects by downregulating HOTAIR through inhibition of this signaling pathway.


**
*TTTY18*
**


GEN suppressed CRC cell growth and migration and promoted apoptosis [186]. GEN downregulated TTTY18 expression and phosphorylation of AKT and p38 MAPK, suggesting that inhibition of TTTY18/AKT pathway is involved in GEN’s anticancer activity.

### 3.5. lncR Modulations by QUE


**
*MALAT1*
**


By data mining including computational analysis, Li et al. [187] identified QUE’s therapeutic candidate genes in cervical cancer HeLa cells. Among them, EGFR, JUN, AR, CD44, and MUC1 were selected, and MALAT1, 10 miRs, and 71 circRs upstream of these genes were determined. These findings lead to the construction of a regulatory network of lncR/circR-miR-mRNA pathway and provided a theoretical basis for targeted therapy of cervical cancer. In PCa PC-3 cells, QUE downregulated the expression of oncogenic MALAT1 and inhibited the growth of these cells and their xenograft tumors [188]. QUE suppressed the EMT process, promoted apoptosis, and downregulated PI3K/AKT signaling pathway. Overexpression of MALAT1 attenuated the QUE’s effects.

QUE treatment decreased the cell viability of HUVEC cells and downregulated the expression of MALAT1 and MIAT [189]. Since MALAT1 is related to endothelial cell growth, metastasis, and angiogenesis and since MIAT regulates angiogenesis through interaction with miR-150-5p, which can target VEGF, QUE may exert its anticancer effects through downregulation of these lncRs.


**
*NEAT1*
**


Sheng et al. [190] found that in a mouse model of acute pancreatitis, QUE downregulated TNF-α, IL-6, and IL-10, while upregulating miR-216b expression, leading to suppression of p38 MAPK signaling pathway. QUE downregulated NEAT1 which is a direct target of miR-216b. NEAT1 was shown to be a direct target of miR-216b and the triad of NEAT1, miR-216b, and MAP2K6 formed a competitive endogenous RNA network. These findings may partially explain QUE’s anticancer effects.


**
*SNHG7*
**


Chai et al. [119] found NSCLC cells had the elevated expression of oncogenic SNHG7 and the decreased expression of miR-34a-5p compared to those in normal cells. QUE downregulated SNHG7 and increased miR-34a-5p levels in these cells. Overexpression of SNHG7 or downregulation of miR-34a-5p promoted NSCLC cell growth and metastasis. The anticancer effects of QUE were counteracted by co-transfection of SNHG7 mimic or miR-34a-5p inhibitor. These results indicate that QUE may exert its anticancer effects by mediating signaling via the SNHG7/miR-34a-5p axis. Based on the previous findings that SNHG7 upregulates AKT/mTOR pathway in NSCLC cells [191], it is plausible that QUE’s downregulation of SNHG7 is related to downregulation of this pathway.


**
*UCA1*
**


In BCa MCF-7 cells, QUE inhibited cell proliferation and induced cell cycle arrest at G2 phase [192]. INXS is a lncRNA that is able to shift the Bcl-X alternative splicing from the anti-apoptotic Bcl-XL to the pro-apoptotic Bcl-XS, and QUE can cause INXS upregulation and UCA1 downregulation in BCa cells, suggesting QUE exerts its anticancer effects through modulation of these lncRs.

### 3.6. lncR Modulations by RES


**
*AK001796*
**


Yang et al. [193] found that AK001796 was overexpressed in lung cancer tissues and cells (A549 and H446) and its expression was downregulated in RES-treated lung cancer cells. Knockdown of AK001796 reduced cell viability and caused a cell cycle arrest at G0/G1.


**
*DLEU2*
**


Kay et al. [194] demonstrated that RES upregulated the tumor suppressor gene DLEU2 in 11 alternative splicing transcripts. Since DLEU2 was shown to negatively regulate cyclins E1 and D1 through upregulation of miR-15a/miR-16-1 and since overexpression of DLEU2 recovered cellular proliferation and inhibition of the colony-forming ability of tumor cells in a miR-15a/miR-16-1-dependent manner [195], RES’s upregulation of DLEU2 may contribute to its anticancer effects.


**
*H19*
**


In GCa SGC7901 cells, 200 µM RES was shown to increase expression of MEG3, PTTG3P and BISPR and decreased expression of GAS5 and H19 [196]. RES at 50 µM upregulated H19 and MALAT1, and knockdown of H19 in RES-treated cells increased the effect of RES on apoptosis, endoplasmic reticulum stress, and cell cycle S-phase arrest in these cells, suggesting that RES increases chemotherapy sensitivity.


**
*MALAT1*
**


RES inhibited invasion and metastasis of CRC LoVo cells and downregulated MALAT1 [197]. RES’s suppressive effects on tumor cell migration and invasion and protein expression of β-catenin, c-Myc, and MMP-7 were attenuated by overexpression of MALAT1. The finding suggests that suppression of Wnt/β-catenin signaling by downregulation of MALAT1 contributes to RES’s anticancer effects.


**
*NEAT1*
**


Geng et al. [198] found higher expression of NEAT1 in multiple myeloma U266 and LP-1 cells compared to normal bone marrow plasmocytes. RES downregulated NEAT1 and counteracted enhanced cell proliferation, migration, and invasion induced by NEAT1 overexpression. NEAT1 overexpression upregulated the expression of nuclear β-catenin, c-Myc, MMP-7 and survivin, leading to activation of the Wnt/β-catenin signaling pathway.


**
*PCAT29*
**


RES upregulated PCAT29 expression and attenuated its downregulation induced by IL-6 [199]. Knockdown of PCAT29 expression increased cell viability, while RES-induced upregulation of PCAT29 resulted in decreased cell viability. Since RES can downregulate IL-6 (Figure 1), RES may exert its anticancer effect by upregulating PCAT29 expression. Since PCAT29 has been reported to upregulate PTEN by downregulation of miR-494 in NSCLC [200], upregulation of PCAT29 can be reasonably related to anticancer activity of RES.


**
*Other studies*
**


In glioma U87 and U251 cells, RES upregulated NEAT1, MIR155HG, MEG3, and ST7OT1 during induction of apoptosis [201]. Since NEAT1 and MIR155HG are oncogenic [202,203,204] and MEG3 is tumor-suppressing [166], the effect of RES on MEG3 may be a predominant contributor to apoptosis of these cells. NEAT1 was demonstrated to activate ERK, which is a component molecule in RSTAPs (Figure 1) [202].

In colon adenocarcinoma HT-29 cells, the results of qRT-PCR indicated that RES decreased the expression of CCAT1, CRNDE, H19, HOTAIR, PCAT1, PVT1, and SNHG16, and upregulated CCAT2, MALAT1, and TUSC7 [205]. Although individual roles of these lncRs in RES’s anticancer effects are not clear, for example, downregulation of HOTAIR may be related to anticancer effects, since better disease-free survival rate was observed in colon adenocarcinoma patients with low HOTAIR expression. It may be considered that RES’s downregulation of CCAT1 contributes to the anticancer effect, because CCAT1 promotes tumor progression by stabilizing PI3K/AKT/mTOR signalling in lung adenocarcinoma [206]. 

### 3.7. Comparison of the Modulation of lncRs by Six Polyphenols

Table 5 shows comparison of the modulation of lncRs by six dietary polyphenols. Only six lncRs (GAS, H19, HOTAIR, MALAT1, NEAT1 and UCA1) have been reported to be modulated similarly by two or three dietary polyphenols (Table 5). In many cases, the modulation lncRs has been studied for one of six polyphenols, but the reason for this is unclear. One possible reason is that lncRs, which have not been studied before, are chosen as targets for study on polyphenol’s effects, because research publications are required to include novel findings. As research progresses, the number of lncRs commonly regulated by six polyphenols with similar chemical properties is likely to increase.

Table 5 also lists determined or possible targets affected by these lncRs. The above discussion and this table indicate that many targets of lncRs are related to RSTAPs, suggesting that modulation of these lncRs by six dietary polyphenols may contribute to their anticancer effects.

## 4. Effects of Dietary Polyphenols on CircRs

CircRs are associated with cancer development by modulating miRNAs involved in cell proliferation, migration, and carcinogenesis [207]. CircRs can reduce the population of active miRs by binding or sponging, thus affecting miR activity like lncRs aforementioned [208]. Currently, limited studies have examined the effects of polyphenols on circRs. Although microarray analysis has shown that six polyphenols affect many circRs [209,210], more research is needed on this issue.


**
*CUR’s downregulation of circ_102115*
**
*, **circ_PRKCA, circ_FNDC3B, and circ_0078710***


Microarray analysis of the radioresistant nasopharyngeal carcinoma cell line CNE-2 and normal cell lines identified 1042 upregulated and 1558 downregulated circRs [210]. CUR was found to confer radiosensitivity to nasopharyngeal carcinoma CNE-2 cells by regulating the circR-miR-mRNA network and inhibiting EGFR signaling, and STAT3, and GRB2. The same group also found that CUR attenuated radiation-induced upregulation of circ_102115 and miR-335-3p, and downregulation of MAPK1. These findings suggest that CUR restores the radiation sensitivity of these cancer cells [207].

Xu et al. [107] found that circ_PRKCA and integrin β1 expression were upregulated in NSCLC tissues and in H460 and A549 cells, while miR-384 was downregulated. Forced expression of integrin β1 expression attenuated the CUR’s inhibitory effect on cell viability and colony formation. CUR promoted apoptosis and reduced migration and invasion of these cells, but these effects were abolished by integrin β1 expression transfection. These results suggest that CUR inhibits cancer cell growth via downregulation of circ_PRKCA, which upregulates integrin β1 expression by adsorbing/sponging miR-384.

In RCa tissues and cells, circ_FNDC3B levels were upregulated and miR-138-5p was downregulated [211]. CUR suppressed cell proliferation and promoted apoptosis in RCa CAKI-1 and ACHN cells. CUR downregulated circ_FNDC3B and upregulated miR-138-5p. These effects were attenuated by overexpression of circ_FNDC3B or knockdown of miR-138-5p. Transfection of circ_FNDC3B attenuated CUR-induced increased apoptosis and increased expression of Bax, while the enhanced expression of cyclin D1 and bcl-2 induced by CUR were restrained. Thus, downregulation of circ_FNDC3B by CUR contributed to enhancement of apoptosis in these cells.

CUR inhibited the proliferation, migration, and invasion of hepatocellular carcinoma (HCC) cell lines HCCLM3 and Huh7 and induced apoptosis [208]. CUR downregulated the expression of circ_0078710 in these cells and siRNA knockdown of circ_0078710 enhanced CUR’s anticancer effects. CUR stimulated miR-378b expression and overexpression of miR-378b enhanced apoptosis in CUR-stimulated HCC cells. DNA primase, PRIM2 expression was reduced by silencing of circ_0078710 and increased by anti-miR378b treatment. circ_0078710 enhanced PRIM2 expression by sponging miR-378b and consequently, CUR’s downregulation of circ_0078710 increased miR378b population, resulting in downregulation of PRIM2 which is involved in cell proliferation.


**
*GEN’s downregulation of circ_0031250*
**


GEN inhibited cell viability and downregulated oncogenic circ_0031250 in NSCLC H292 and A549 cells [140]. Circ_0031250 knockdown using siRNA further increased the expression of Bax protein, which was upregulated by GEN. Circ_0031250 knockdown also caused enhanced apoptosis, which was mitigated by the miR-873-5p inhibitor in the presence of GEN. miR-873-5p overexpression increased Bax expression in GEN-treated cells, which was restored by FOXM1 upregulation. Thus, GEN exerts its anticancer effect on these cells by modulating circ_0031250/miR-873-5p/FOXM1 axis.


**
*QUE’s upregulation of circ_R 8:93786223|93822563*
**


In CRC HCT 116 cells, QUE inhibited cell proliferation and induced apoptosis. Zhang et al. [209] examined miRNA, lncR, circR, and mRNA expression profiles and found QUE-induced differential expression in 131 circRs, 240 lncRs, 83 miRs, and 1415 mRNAs in these cells. QUE upregulated the expression of miRs including miR-5096, and also circR 2:206841107|206881891 and circR 2:206866697|206881891. Furthermore, QUE decreased mRNA levels including LRG1 mRNA and lncRs including cirR 8:93786223|93822563. Binding site analysis indicated that circR 8:93786223|93822563 can interact with LRG1 mRNA through competitive binding with miR-5096. A previous report showed a strong association of LRG1 with worse overall survival in CRC. Thus, the interactive relationships of circR 8:93786223|93822563–miR-5096–LRG1 may be involved in the mechanism of action of QUE, but it appears difficult to clarify this mechanism, because QUE affects the expression of so many circRs, lncRs, miRs, and mRNAs.

## 5. Conclusions

Anticancer effects of dietary polyphenols, CGA, CUR, EGCG, GEN, QUE, and RES have been shown in many human, animal, and cell-based studies. These results may explain those of epidemiological studies demonstrating the reduced cancer risk of a variety of cancer types by consumption of coffee, green tea, soybeans, wine, and curry. A cohort study conducted by Wang et al. reported that the amount of total polyphenols ranged from 8.88 to 47.44 mg/day in the lowest and highest quintile, respectively, among 38,408 middle-aged and old women [212]. Several studies have shown a correlation between polyphenols-rich diets and a reduced risk of cancer [213,214]. These polyphenols have similar properties in that they can act as antioxidants to scavenge ROS, which triggers pathways leading to cell cycle arrest, apoptosis induction, anti-inflammation, and anti-angiogenesis.

These polyphenols can modulate the expression/activity of several RSTAP components by increasing the expression of tumor-suppressive ncRNAs and decreasing the expression of oncogenic ncRNAs in general. Thus, modulation of ncRNAs may additively or synergistically enhance the anticancer effects of these polyphenols via RSTAPs.

Although six dietary polyphenols have similar chemical properties, GEN and RES, and EGCG can act uniquely as phytoestrogens and a 67kDa laminin receptor, respectively, in anticancer effects [7,113,215]. These effects together with different modulation found among six dietary polyphenols on ncRNAs may explain differences in cancer-specific effects of consumption of foods containing these polyphenols observed in epidemiological studies [7,8], but more precise analysis is needed to clarify this issue.

In a recent comprehensive review, Fujimura et al. [216] have discussed that EGCG sensing by the 67kDa laminin receptor can be potentiated by a variety of biomolecules such as citrus polyphenols and sulfur-containing food factors. Therefore, it should be interesting to search cellular sensing systems for other dietary polyphenols which will lead to lower their active concentrations and examine involvements of ncRNAs in the anticancer mechanism.

Since this review is written based on the data from two databases: PubMed (https://pubmed.ncbi.nlm.nih.gov/ (accessed on 24 November 2022)) and Web of Science (http://webofknowledge.com/WOS (accessed on 24 November 2022)), there is a limitation that this review has not covered all of findings related to the present theme.

## Figures and Tables

**Figure 1 antioxidants-11-02352-f001:**
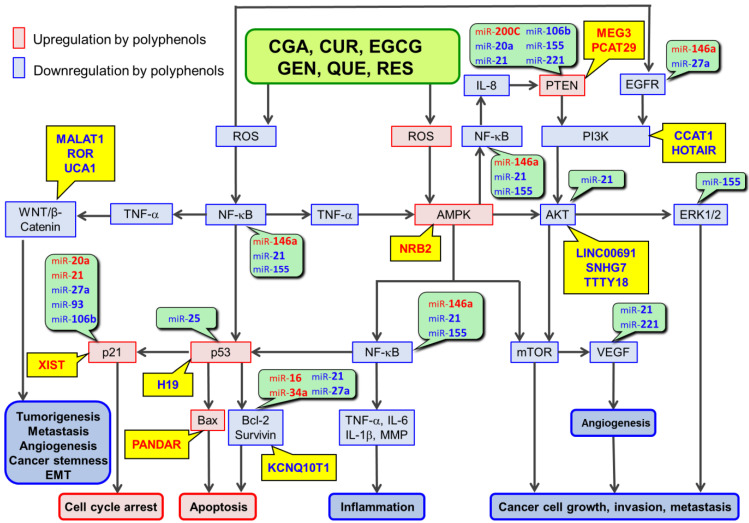
ROS-scavenger-triggered anticancer pathways (RSTAPs) and contribution of ncRNAs. lncRs upregulated and downregulated by polyphenols are in red and blue, respectively, on a yellow background. miRs upregulated and downregulated by polyphenols are in red and blue, respectively, on a green background.

**Table 1 antioxidants-11-02352-t001:** Modulation of molecular targets of tumor-suppressor miRs upregulated by three to five CUR, EGCG, QUE, RES and GEN.

miRs	miR-16	miR-22	miR-34a	miR-141	miR-145	miR-146a	miR-200c
**Polyphenols**	**CUR** Yang et al. [9] **EGCG** Tsang et al. [10] **QUE** Sonoki et al. [11]; Zhao et al. [12] **RES** Hagiwara et al. [13]; Azimi et al. [14]	**CUR** Sun et al. [15]; Sreenivasan et al. [16]; Sibbesen et al. [17] **EGCG** Li et al. [18] **QUE** Zhang et al. [19]	**CUR** Guo et al. [20]; Sun et al. [21]; Toden et al. [22]; Sun et al. [15] **EGCG** Chakrabarti et al. [23]; Li et al. [18]; Chakrabarti et al. [24]; Toden et al. [25]; Mostafa et al. [26] **GEN** Hsieh et al. [27]; Xia et al. [28]; Chiyomaru et al. [29] **RES** Hagiwara et al. [13]; Otsuka et al. [30]; Kumazaki et al. [31]; Yao et al. [32]	**CUR** Toden et al. [33] **EGCG** Gordon et al. [34] **GEN** Chiyomaru et al. [35] **RES** Hagiwara et al. [13]	**CUR** Mirgani et al. [36]; Liu et al. [37] **EGCG** Toden et al. [25] **GEN** Wei et al. [38] **QUE** Zhou et al. [39] **RES** Sachdeva et al. [40]	**CUR** Wu et al. [41] **GEN** Li et al. [42] **QUE** Tao et al. [43]	**CUR** Toden et al. [33]; Soubani et al. [44] **EGCG** Toden et al. [25] **RES** Hagiwara et al. [13]; Dermani et al. [45]
**Targets ***	Bcl-2↓ **CUR**: Yang et al. [9]; **EGCG**: Tsang et al. [10] HOXA10↓ **QUE**: Zhao et al. [12]	SP1↓, ESR1↓ **CUR**: Sun et al. [15] Erbb3↓ **CUR**: Sreenivasan et al. [16] NCoA1↓, HDAC6↓, MYCBP↓, PTEN↓, **CUR**: Sibbesen et al. [17] Wnt1/β-catenin↓ **QUE**: Zhang et al. [19]	Bcl-2↓, Bmi-1↓ **CUR**: Guo et al. [20] Bcl-2↓, CDK4↓, Cyclin D1↓ **CUR**: Sun et al. [21] Cyclin D↓, c-Myc↓, CDK6↓, Bcl-2↓ **CUR**: Toden et al. [22] miR-92↓, miR-93↓, miR-106b↓, miR-7-1↑, miR-34a↑, miR-99a↑ **EGCG**: Chakrabarti et al. [23] EMT↓, RTCB↓, ROS↑ **GEN**: Hsieh et al. [27] HOTAIR↓ **GEN**: Chiyomaru et al. [29] Notch-1↓ **GEN**: Xia et al. [28] Sirt1↓ via E2F3 **RES**: Kumazaki et al. [31] HNRNPA1↓ **RES**: Otsuka et al. [30] Bcl-2↓ **RES**: Yao et al. [32]	EMT↓ **CUR**: Toden et al. [33] HOTAIR↓ **GEN**: Chiyomaru et al. [35] Cancer stemness↓ **RES**: Hagiwara et al. [13]	Oct4↓, SOX-2↓, Oct4B1↓, **CUR**: Mirgani et al. [36] Oct4↓, CD44↓, CD133↓, Cyclin D1↓, Cdk4↓ **CUR**: Liu et al. [37] c-Myc↓ **EGCG**: Toden et al. [25] ABCE1↓ **GEN**: Wei et al. [38] Caspase-3↑ **QUE**: Zhou et al. [39]	NF-κB↓ **CUR**: Wu et al. [41] EGFR↓ MTA-2↓, IRAK-1↓, NF-κB↓ **GEN** Li et al. [42] Caspase 3↑, Bax↑, EGFR↓ **QUE**: Tao et al. [43]	EMT↓ **CUR**: Toden et al. [33] PTEN↑, MT1-MMP↓ **CUR**: Soubani et al. [44] Cancer stemness↓ **EGCG**: Toden et al. [25] Cancer stemness↓ **RES**: Hagiwara et al. [13] EMT↓ via vimentin, ZEB1, E-cadherin **RES**: Dermani et al. [45]

* Targets upregulated and downregulated by polyphenols through upregulation of miRs are indicated by ↑ and ↓, respectively. Bcl-2; B-cell lymphoma 2, HOXA10; homeobox A10, SP1; specificity protein 1, ESR1; estrogen receptor alpha 1, Erbb3; Erb-b2 receptor tyrosine kinase 3, NCoA1; nuclear receptor coactivator 1, HDAC6; histone deacetylase 6, MYCBP; Myc binding protein, PTEN; phosphatase and tensin homolog deleted on chromosome 10, Wnt1; wingless and int-1, EMT; epithelial-mesenchymal transition, RTCB; RNA 2′,3′-cyclic phosphate and 5′-OH ligase, ROS; reactive oxygen species, HOTAIR; HOX transcript antisense RNA, Notch-1; Notch homolog protein 1, Sirt1; sirtuin 1, ABCE1; ATP-binding cassette E1, NF-κB; nuclear factor-κB, EGFR; epidermal growth factor receptor, MTA-2; metastasis-associated 2, Bax; Bcl2-associated X protein, MT1-MMP; membrane type 1-matrix metalloproteinase, Bmi-1; B-cell-specific Moloney murine leukemia virus integration site 1, CDK; cyclin-dependent kinase, E2F3; E2F transcription factor 3, HNRNPA1; heterogeneous nuclear ribonucleoprotein A1, Oct4; octamer-binding transcription factor 4, SOX-2; SRY [sex determining region Y]-box 2, ZEB1; zinc finger E-box binding homeobox 1.

**Table 2 antioxidants-11-02352-t002:** Modulation of molecular targets of oncogenic miRs downregulated by three to five CUR, EGCG, QUE, RES and GEN.

miRs	miR-20a	miR-21	miR-25	miR-27a	miR-93	miR-106b	miR-155	miR-221
**Polyphenols**	**CGA** Huang et al. [46] **CUR** Gandhy et al. [47] **EGCG** Mirzaaghaei et al. [48] **RES** Dhar et al. [49]; Dhar et al. [50]	**CGA** Wang et al. [51] **CUR** Mudduluru et al. [52]; Subramaniam et al. [53]; Zhang et al. [54]; Taverna et al. [55]; Yallapu et al. [56] **EGCG** Fix et al. [57] **; Siddiqui et al. [58] **GEN** Zaman et al. [59] **RES** Tili et al. [60]; Sheth et al. [61]; Liu et al. [62]; Li et al. [63]; Zhou et al. [64]	**CUR** Sun et al. [15] **EGCG** Fix et al. [57] **; Gordon et al. [34]; Zan et al. [65] **RES** Tili et al. [60]	**CUR** Toden et al. [22]; Noratto et al. [66] **EGCG** Fix et al. [57] ** **GEN** Xia et al. [67]; Xu et al. [68]; Sun et al. [69]	**CGA** Huang et al. [46] **EGCG** Chakrabarti et al. [23]; Chakrabarti et al. [24] **RES** Singh et al. [70]	**CGA** Huang et al. [46] **EGCG** Chakrabarti et al. [23] **RES** Dhar et al. [50]; Dhar et al. [49]	**CGA** Zeng et al. [71] **CUR** Ma et al. [72] **GEN** de la Parra et al. [73] **QUE** Boesch-Saadatmandi et al. [74] **RES** Tili et al. [75]	**CUR** Zhang et al. [76]; Allegri et al. [77] **EGCG** Allegri et al. [77] **GEN** Chen et al. [78]
**Targets ***	p21↑ **CGA**: Huang et al. [46] PTEN↑ **RES**: Dhar et al. [49]	Smad7↑ **CGA**: Wang et al. [51] PDCD4↑ **CUR**: Mudduluru et al. [52] PTEN↑ **CUR**: Zhang et al. [54] PTEN↑ **CUR**: Taverna et al. [55] p21↑, p38 MAPK↑, Cyclin E2↓ **GEN**: Zaman et al. [59] PDCD4↑ **RES**: Sheth et al. [61] Bcl-2↓ **RES**: Liu et al. [62] NF-κB↓ **RES**: Liu et al. [63] AKT↓, Bcl-2↓ **RES**: Zhou et al. [64]	p53↑ **EGCG**: Gordon et al. [34] PARP1↑, Caspases 3↑, Caspases 9↑ **EGCG**: Zan et al. [65]	Cyclin E1↓, c-Myc↓ via FBXW7 **CUR**: Toden et al. [22] ZBTB10-Sp↑ **CUR**: Noratto et al. [66] Sp1↓, Sp3↓ Sp4↓, EGFR↓, hepatocyte growth factor receptor↓, survivin↓, Bcl-2↓, Cyclin D1↓, NFκB↓, ZBTB4↑ **CUR**: Grandhy et al. [47] Spry2↑ **GEN**: Xu et al. [68]	p21↑ **CGA**: Huang et al. [46] Caspase 8↑, tBid↑, Calpain↑, Caspase 3↑ **EGCG**: Chakrabarti et al. [23]	p21↑ **CGA**: Huang et al. [46] PTEN↑ **RES**: Dhar et al. [50]; Dhar et al. [49]	Inflammation↓ via NF-κB/NLRP3 **CGA**: Zeng et al. [71] SOCS1↓, IL-6↓, **CUR**: Ma et al. [72] PTEN↑, FOXO3a↑ **GEN**: de la Parra et al. [73] AP-1↓ via miR-663 **RES**: Tili et al. [75]	PTEN↑, p27↑, p57↑, PUMA↑ **CUR**: Sarkar et al. [79] FGF2↓, MMP2↓, VEGF↓, HGF↓, **CUR**: Zhang et al. [76] miR-21↓, miR-146b↓, miR-221↓, miR-222↓ **CUR**: Allegri et al. [77] miR-221↓, **EGCG**: Allegri et al. [77] ARHI↑ **GEN**:Chen et al. [78]

* Targets upregulated and downregulated by polyphenols through downregulation of miRs are indicated by ↑ and ↓, respectively. ** EGCG-rich Polyphenon-E. PTEN; phosphatase and tensin homolog deleted on chromosome 10, Smad; Small mother against decapentaplegic, PDCD; programmed cell death, Bcl-2; B-cell lymphoma 2, NF-κB; nuclear factor-κB, AKT; AKT serine/threonine kinase 1, PARP; poly(ADP-ribose) polymerase 1, FBXW7; F-Box And WD Repeat Domain Containing 7, ZBTB10; Zinc finger and BTB domain containing 10, Sp1; specificity protein 1, EGFR; epidermal growth factor receptor, Spry2; Sprouty RTK signaling antagonist 2, tBid; truncated BH3 interacting domain death agonist, NLRP3; NLR family, pyrin domain containing 3, SOCS1; suppressor of cytokine signal 1, IL-6; interleukin-6, FOXO3a; forkhead Box O3, AP-1; Activator protein 1, PUMA; p53-upregulated modulator of apoptosis, ARHI; age-related hearing impairment, FGF2; fibroblast growth factor 2, MMP; matrix metalloproteinase, VEGF; vascular endothelial growth factor, HGF; hepatocyte growth factor.

**Table 3 antioxidants-11-02352-t003:** miRs upregulated by one of five dietary polyphenols and their proposed targets *.

CUR		EGCG	GEN	QUE	RES
miR-7 SET8↓, Bcl-2↓, p53↑ [80]; Skp2↓, p57↑, p21↑ [81] miR-9 AKT↓, FOXO1↓ [82]; GSK-3β↑, β-catenin↑, Cyclin D1↓ [83] miR-15a Bcl-2↓ [9]; WT1↓ [84] miR-16-1 WT1↓ [84] miR-28-5p BECN1↓ [85] miR-29a DNMT1↓, 3A↓, 3B↓ [86] miR-30c-5p MTA1↓ [87] miR-33b HMGA2↓ [88]; XIAP↓ [89] miR-98 LIN28A↓, MMP2↓, MMP9↓ [90] miR-99a JAK1↓, STAT1↓, STAT3↓ [91] miR-101 EZH2↓, EpCAM↓ [92]; Notch1↓ [93]; EZH2↓ [94] miR-124 Midkine↓ [95]	miR-125a ERRα↓ [96] miR-138 Smad4↓, NF-kB↓, Cyclin D3↓ [97] miR-143 NF-kB↓ [98]; PGK1↓ [99]; Autophagy via ATG2B↓ [100] miR-181b CXCL1↓ [101] miR-185 DNMT1↓, 3A↓, 3B↓ [86] miR-192-5p XIAP↓ [102]; PI3K↓, AKT↓ [103]; Wnt/β-catenin↓ [104] miR-196b ** BCR-ABL↓ [55] miR-206 mTOR↓, AKT↓ [105] miR-215 XIAP↓ [102] miR-340 XIAP↓ [106] miR-384 circ-PRKCA↓ [107] miR-491 PEG10↓ [108] miR-593 MDR1↓ [109]	miR-15b STIM2↓, Orai1↓ [110] miR-29b KDM2A↓ [111] miR-485-5p RXRα↓ [112] let-7b HMGA2↓ [113]	miR-574-3p RAC1↓, EGFR↓, EP300↓ [114] miR-1469 Mcl1↓ [115] let-7d THBS1↓ [116]	miR-1-3p TAGLN2↓ [117] miR-16-5p WEE1↓ [118] miR-22 Wnt1↓ [19] miR-34a-5p SNHG7↓ [119] miR-142-3p HSP70 ↓ [120] miR-197 IGFBP5↓ [121] miR-200b-3p Notch1↓ [122] miR-217 KRAS↓ [123] miR-503-5p Cyclin D1↓ [124] miR-1254 CD36↓ [125] miR-1275 IGF2BP1↓, IGF2BP3↓ [126] let7-a KRAS↓ [127] let-7c Numbl/Notch1↑ [128]	miR-424-3p Galectin-3↓ [129]

* Upregulation (↑) and downregulation (↓) of miR targets by polyphenols are indicated. ** Downregulation by RES is reported [130]. SET8; SET domain-containing lysine methyltransferase 8, Bcl-2; B-cell lymphoma 2, Skp2; S-phase kinase-associated protein 2, AKT; AKT serine/threonine kinase 1, FOXO1; forkhead Box O1, GSK-3β; glycogen synthase kinase-3 beta, WT1; Wilms’ tumor-1, BECN1; beclin 1, DNMT; DNA methyltransferase, MTA1; metastasis-associated 1, HMGA2; high mobility group A2, XIAP; X-linked inhibitor of apoptosis, LIN28A; Lin-28 homolog A, MMP; matrix metalloproteinase, JAK1; Janus kinase 1; STAT; signal transducer and activator of transcription, EZH2; enhancer of zeste homolog 2, EpCAM; epithelial cell adhesion molecule, Notch1; neurogenic locus notch homolog protein 1, ERRα; estrogen-related receptor alpha, PGK1; phosphoglycerate kinase 1, ATG2B; autophagy-related 2B, CXCL1; chemokine (C-X-C motif) ligand 1, PI3K; phosphoinositide-3 kinase, Wnt; wingless and int-1, BCR-ABL; BCR-ABL fusion gene, mTOR; mammalian target of rapamycin, circ-PRKCA; circ_0007580, PEG10; paternally expressed gene 10, MDR1; multidrug resistance mutation1, STIM2; Stromal interaction molecule 2, Orai1; ORAI calcium release-activated calcium modulator 1, KDM2A; lysine demethylase 2A, RXRα; retinoid X receptor alpha, RAC1; ras-related C3 botulinum toxin substrate 1, EGFR; epidermal growth factor receptor, EP300; E1A-associated protein P300, THBS1; thrombospondin 1, TAGLN2; transgelin 2, WEE1; WEE1 G2 checkpoint kinase, SNHG7, small nucleolar RNA host gene 7, HSP; heat shock protein, IGFBP; insulin-like growth factor binding protein, KRAS; KRAS proto-oncogene, GTPase, Numbl; NUMB like endocytic adaptor protein, Mcl1; myeloid cell leukemia 1, IGF2BP; insulin-like growth factor 2 mRNA binding protein.

**Table 4 antioxidants-11-02352-t004:** miRs downregulated by one of six dietary polyphenols and their targets *.

CGA	CUR	EGCG	GEN	QUE	RES
miR-17 p21↑, G0/G1 arrest↑ [46]	miR-19a,b PTEN↑ [131] miR-125a-5p TP53↑ [132] miR-130a Nkd2↑ [133] miR-7641 p16↑ [134]	miR-98-5p CTR1↑ [135]	miR-23b-3p PTEN↑ [136] miR-151a-5p CASZ1↑, IL1RAPL1↑, SOX17↑, N4BP1↑, ARHGDIA↑ [137] miR-155 PTEN↑ [73] miR-221 miR-222 ARHI↑ [78] miR-223 Fbw7↑ [138] miR-223 E-cadherin↑ [139] miR-873-5p FOXM1↓ [140] miR-1260b sFRP1↑, Smad4↑, Dkk2↑ [141,142]	miR-30d-5p Notch↓ Wnt↓ [143]	miR-196b ** miR-1290 IGFBP3↑ [130]

* Upregulation (↑) and downregulation (↓) of miR targets by polyphenols are indicated. ** Upregulation by CUR is also reported [55]. PTEN; phosphatase and tensin homolog deleted on chromosome 10, Nkd2; naked cuticle homolog 2, CTR1; copper transporter 1, CASZ1; castor zinc finger 1, IL1RAPL1; interleukin 1 receptor accessory protein like 1, SOX17, SRY-box transcription factor 17, N4BP1; NEDD4-binding protein 1, ARHGDIA, rho-GDP dissociation inhibitor-alpha, ARHI; age-related hearing impairment, Fbw7; F-Box and WD repeat domain-containing 7, FOXM1; forkhead box M1, sFRP1; secreted frizzled-related protein 1, Smad; small mother against decapentaplegic, Dkk2; Dickkopf-related protein 2, Wnt; wingless and int-1, IGFBP; insulin-like growth factor binding protein.

**Table 5 antioxidants-11-02352-t005:** Modulation of lncRs by six dietary polyphenols and targets affected by lncRs.

lncR	Upregulation	Downregulation	Effects of Polyphenols on Proposed Targets of lncRs (↑, Upregulation; ↓, Downregulation)
AF085935		EGCG [177]	Not specified
AK001796		RES [193]	Cell-cycle arrest↑ [193]
AK294004		CUR [156]	Cyclin D1↓ [156]
CCAT1		RES [205]	PI3K/AKT/mTOR↓ [206]
DLEU2	RES [194]		Cyclins E1 and D1↓ [195]
GAS5	CGA [152] CUR [157]		miR-23a↓ by sponging [152]
H19	RES (at 50 μM) [196]	CUR [159,160,161] RES (at 200 μM) [162]	Cell cycle arrest at S-phase↑ [196], p53↑ [160] EMT↓, invasion and migration via upregulating Snail [161]
HOTAIR		GEN [29,35,182,183] RES, [205]	PI3K/AKT/mTOR signaling pathway↓ [184,185]
KCNQ1OT1		CUR [162]	Bcl-2↓ via miR-497 [162]
LINC00511		EGCG [111]	miR-29b↑ [111]
LINC00691		CUR [163]	AKT↓ [163]
LINC-PINT	CUR [164]		Cell cycle arrest at G2/M↑ [164]
MALAT1	RES [196]	QUE [187,188,189] RES [197]	Wnt/β-catenin signaling↓ [197]
MEG3	CUR [165,167]		PTEN↑ [167]
NEAT1	EGCG [135,178]	EGCG [179] QUE [190] RES [198]	ERK1/2↓ [178] miR-98-5p↑ [135] by sponging Cancer cell stemness↓ [179]
NRB2	CUR [168]		AMPK/mTOR pathway↑ [168]
PANDAR	CUR [169]		Bax↑ via upregulation of PUMA [169,170]
PCAT29	RES [199]		PTEN↑ via downregulation of miR-494 [200]
ROR		CUR [37,172]	Wnt/β-catenin↓ [172] Oct4↓ via sponging miR-145 [37]
SNHG7		QUE [119]	AKT/mTOR↓ [191]
SOX2OT variant 7		EGCG [180]	Cancer cell stemness↓ in combination with doxorubicin [180]
TTTY18		GEN [186]	AKT↓ [186]
TUG1	CGA [153]		Not specified
UCA1		CUR [174] QUE [192]	Wnt/mTOR↓ [174] c-Myc↓ by sponging miR-124 [175]
XIST	CUR [176]		p21↑ [176]

PI3K; phosphatidylinositol 3-kinase, AKT; AKT serine/threonine kinase 1, mTOR; mammalian target of rapamycin, Bcl-2; B-cell lymphoma 2, PTEN; phosphatase and tensin homologs deleted on chromosome 10, ERK; extracellular signal-regulated kinase, AMPK; AMP-activated protein kinase, Bax; Bcl2-associated X protein, oct4; Octamer-binding transcription factor.

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
