# Peer review of "Contribution of Non-Coding RNAs to Anticancer Effects of Dietary Polyphenols: Chlorogenic Acid, Curcumin, Epigallocatechin-3-Gallate, Genistein, Quercetin and Resveratrol"

_antioxidants, 2022, doi:10.3390/antiox11122352_

Round 1

Reviewer 1 Report

Growing evidence has been accumulated to show the anticancer effects of daily consumption of polyphenols. Polyphenols can act as antioxidants and exert the anticancer effects via anticancer pathways involving e.g. their reactive oxygen species scavenging activity. n this review Authors discussed the miR-modulating effects of polyphenols, which have been reported in studies using one or two of the 6 major dietary polyphenols (chlorogenic acid, curcumin, epigallocatechin-3-O-gallate, genistein, quercetin, and resveratrol). The review raises interesting issues, however, it has not been presented in a lucid manner. The manuscript requires additions and corrections before being published in Antioxidants.

Specific comments:

1) The article is drafted chaotic and neglectfuly prepared, the tables should be corrected as required by the journal, the table at the end does not contain all abbreviations etc.

2) Figure 1 is of poor quality and should be corrected or removed - in my opinion structural formulas of these polyphenols are commonly known and unnecessary in this publication.

3) Authors should clearly state  throughout the text and in tables which studies are human-, animal- and cell-based.

4) In general, Authors should clearly define the purpose of the review and stick to the topic chosen. 

Keeping in mind the following as a key message would be certainly useful:

"Reviewing the literature is not stamp collecting. A good review does not just summarize the literature, but discusses it critically, identifies methodological problems, and points out research gaps. After having read a review of the literature, a reader should have a rough idea of:

i. the major achievements in the reviewed field,

ii. the main areas of debate, and

iii. the outstanding research questions."

Reviewer 2 Report

The paper deal with interesting topic

The topic is somehow new and can be of interest for different readers and basis for new or going on researches as being a general review

As presented the paper for me suffer of two main weakness.

First authors need to clarify the novelty of the present review compared to similar and to other they also cite in the introduction. What is the improvement in this specific paper. May this can be better explained.

Second I feel that as organised the work is informative but very difficult to read. There is (and I understand and know that is quite not evitable) a large and extensive use of acronime and abbreviations. But the authors may include for each subchapter a small explanation of the target or the main importance of each target to help also the less expert readers to enter easily in the paper topic.

Second more general point the conclusion need to be changed and also in my opinion some more references in the introduction related to the polyphenols can be implemented

A second general point is that in my humble opinion, there is somehow some confusion (or can be generated in some readers) related to anticancer as therapeutic or preventive agents. So please try to be clear.
Final point rational for the choice of the polyphenol should be better explained in the introduction

A general conclusion considering also the amount and the frequency of the intake of such compounds must be included to understand the real impact of these compounds as cancer preventives.

Some minor comments are also indclude in the pdf

Reviewer 3 Report

I read with a great interest the review "Contribution of non-coding RNAs to anticancer effects of dietary polyphenols: chlorogenic acid, curcumin, epigallocatechin-3-gallate, genistein, quercetin and resveratrol" by Sumio Hayakawa et al. The authors summarized the knowledge about anticancer effects of six most important dietary polyphenols , focusing on epigenetic modifications through various types of noncoding RNAs such as miRNAs, long noncoding RNAs and circular RNAs. The topic is very interesting and very broad.

A large number of abbreviations (a wide variety of noncoding RNAs, many intracellular molecules, tumor types and cell lines) make it difficult for the reader to focus on the content. However, with this type of review, a greater simplification of the text is probably not possible.

Reviewer 4 Report

In the submitted manuscript Hayakawa et al. gave an overview on the contribution of non-coding RNAs (microRNAs, lncRNAs and circRNAs) to anticancer effects of six dietary polyphenols.

This review manuscript is extensive and comprehensive, all available literature on that topic have been included, and the text has been clearly presented and is easily readable.

However, to avoid any ambiguities, some additional data must be provided:

1) Most of the time authors just provided symbols of stem-loop miRNAs, however the mature miRNAs -5p and -3p, if both exist, are not synonyms, i.e., they are not targeting same genes! Therefore, if available in the original literature, authors should always write symbol of the mature miRNA (-5p or -3p).

2) Authors most of the time (correctly) wrote precise names of the cell lines which were used in cited study. However, at some points this information is missing and should be added: lines 139, 208, 214, 264, 390, 471, 493.

3) Line 249: replace "sponge RNA" with "miR sponge"

4) Lines 311-314: Provide lncRNA symbols instead of names of Ensembl genes (ENSG...).

5) Lines 450-451: Rephrase sentence "regulation of lncRs has been studied in one polyphenol of 6 MDPs" because it is incomprehensible how lncRNA could be studied "in polyphenol"?!

6) Consistently write "MDPs" when you mean more than one MDP. Also, numbers less than 10 should be written as words.

7) Lines 548-549: Provide web-addresses for those databases.

8) It seems that something is missing in Figure 2 legend. Should there be a full stop after word "ncRNAs"?

Round 2

Reviewer 1 Report

The Authors revised the manuscript according to the comments of the Reviewers. In my opinion, the article is suitable for publication in this form.

Author Response

Thank you very much for your positive comments on our paper.